# Assessing Privacy Risks in Language Models: A Case Study on Summarization Tasks

**Ruixiang Tang**♣   **Gord Lueck**◇   **Rodolfo Quispe**◇   **Huseyin A Inan**♠
**Janardhan Kulkarni**♠   **Xia Hu**♣
♣Department of Computer Science, Rice University, TX, USA
◇Microsoft, Redmond, WA, USA
♠Microsoft Research, Redmond, WA, USA
{rt39,xia.hu}@rice.edu {gordonl,edquispe,huseyin.inan,jakul}@microsoft.com

## Abstract

Large language models have revolutionized the field of NLP by achieving state-of-the-art performance on various tasks. However, there is a concern that these models may disclose information in the training data. In this study, we focus on the summarization task and investigate the membership inference (MI) attack: given a sample and black-box access to a model's API, it is possible to determine if the sample was part of the training data. We exploit text similarity and the model's resistance to document modifications as potential MI signals and evaluate their effectiveness on widely used datasets. Our results demonstrate that summarization models are at risk of exposing data membership, even in cases where the reference summary is not available. Furthermore, we discuss several safeguards for training summarization models to protect against MI attacks and discuss the inherent trade-off between privacy and utility.

## 1 Introduction

Text summarization seeks to condense input document(s) into a shorter, more concise version while preserving important information. The recent large language models have significantly enhanced the quality of the generated summaries (Rothe et al., 2021; El-Kassas et al., 2021; Chung et al., 2022). These models have been applied to many sensitive data, such as clinical and finance reports (Zhang et al., 2020b; Abacha et al., 2021). Given these high-stakes applications, it is critical to guarantee that such models do not inadvertently disclose any information from the training data and that data remains visible only to the client who owns it.

To evaluate the potential memorization of specific data by a model, *membership inference* (MI) attack (Shokri et al., 2017) has become the de facto standard, owing to its simplicity (Murakonda et al., 2021). Given a model and sample (input label pair), the membership inference attack wants to identify whether this sample was in the model's training

dataset. This problem can be formulated as an adversarial scenario, with Bob acting as the attacker and Alice as the defender. Bob proposes methods to infer the membership, while Alice attempts to make membership indistinguishable. In the past, researchers proposed various attacking and defending techniques. However, most of them focus on the computer vision classification problem with a fixed set of labels. Little attention has been given to comprehending MI attacks in Seq2Seq models.

This paper focuses on the summarization task and investigates the privacy risk under membership inference attacks. Inspired by previous research in MI literature(Shokri et al., 2017; Hisamoto et al., 2020), we pose the problem as follows: *given black-box access to a summarization model's API, can we identify whether a document-summary pair was used to train the model?*. Compared to membership inference attacks on fixed-label classification tasks, text generation tasks present two significant challenges: (1) The process of generating summaries involves a sequence of classification predictions with variable lengths, resulting in complex output space. (2) Existing attacks heavily rely on the output probabilities (Shokri et al., 2017; Mireshghallah et al., 2022), which is impractical when utilizing APIs of Seq2Seq models. Therefore, it remains uncertain whether the methodologies and findings developed for classification models can be applied to language generation models.

A pertinent question emerging from the study is the rationale behind the efficacy of MI attacks on summarization models. The key insight lies in the training objective of these models, which aims to minimize the discrepancy between the generated and reference summary (See et al., 2017). Consequently, samples with significantly lower loss (indicating a similarity between the generated and reference summaries) are more likely to be part of the training dataset. Based on this concept, we propose a baseline attack method that utilizes the similarity

between the generated and reference summaries to differentiate between training and non-training samples. One limitation of the baseline attack is that Bob requires access to both the documents and reference summaries to launch the attack, rendering the attack less practical for summarization tasks.

Building upon this, our study introduces a more general document-only attack: *Given only a document and black-box access to the target model's APIs, can Bob infer the membership without access to the reference summary?* We tackle this problem by examining the robustness of generated summaries in response to perturbations in the input documents. According to the max-margin principle, training data tends to reside further away from the decision boundary, thus exhibiting greater resilience to perturbations, which aligns with observations from prior research in the adversarial domain (Tanay and Griffin, 2016; Choquette-Choo et al., 2021). Consequently, Bob can extract fine-grained membership inference signals by evaluating data robustness toward perturbations. Remarkably, we show that Bob can estimate the robustness without reference summaries, thereby enabling a document-only attack. In summary, this work makes the following contributions to the language model privacy:

1. Defined the black-box MI attack for the sequence-to-sequence model. Experiments on summarization tasks show attackers can reliably infer the membership for specific instances.

2. Explored data robustness for MI attacks and found that the proposed approach enables attackers to launch the attack solely with the input document.

3. Evaluate factors impacting MI attacks, such as dataset size, model architectures, etc. We also explore multiple defense techniques and discuss the privacy-utility trade-off.

## 2 Background and Related Works

**Membership Inference Attacks.** In a typical black-box MI attack scenario, as per the literature (Shokri et al., 2017; Hisamoto et al., 2020), it is posited that the attacker, Bob, can access a data distribution identical to Alice's training data. This access allows Bob to train a shadow model, using the known data membership of this model as ground truth labels to train an attack classifier. Bob can then initiate the attack by sending queries to Alice's model APIs. Most previous studies leverage disparities in prediction distributions to distinguish between training and non-training samples. However, this approach is not feasible for Seq2Seq models. For each generated token in these models, the output probability over the word vocabulary often comprises tens of thousands of elements—for instance, e.g., the vocabulary size for BART is 50,265 (Lewis et al., 2020). As such, most public APIs do not offer probability vectors for each token but rather furnish an overall confidence score for the sequence, calculated based on the product of the predicted tokens' probabilities.

**Natural Language Privacy.** An increasing body of work has been conducted on understanding privacy risk in NLP domain,(Hayes et al., 2017; Meehan et al., 2022; Chen et al., 2022; Ponomareva et al., 2022). Pioneering research has been dedicated to studying MI attacks in NLP models. The study by (Hisamoto et al., 2020) examines the black-box membership inference problem of machine translation models. They assume Bob can access both the input document and translated text and use BLEU scores as the membership signal, which is similar to our baseline attack. (Song and Shmatikov, 2019) investigate a white-box MI attack for language models, which assume Bob can obtain the probability distribution of the generated token. Different from previous work, our attack is under the black-box setting and considers a more general document-only attack in which Bob only needs input documents for membership inference.

## 3 Preliminaries

### 3.1 Problem Definition

We introduce two characters, Alice and Bob, in the membership inference attack problem.

**Alice (Defender)** trains a summarization model on a private dataset. We denote a document as $f$ and its corresponding reference summary as $s$. Alice provides an API to users, which takes a document $f$ as input and returns a generated summary $\hat{s}$.

**Bob (Attacker)** has access to data similar to Alice's data distribution and wants to build a binary classifier $g(\cdot)$ to identify whether a sample is in Alice's training data, $A_{train}$. The sample comprises a document $f$ and its reference summary $s$. Together with the API's output $\hat{s}$, Bob uses $g(\cdot)$ to infer the membership, whose goal is to predict:

$$g(f, s, \hat{s}) = \begin{cases} 1 & \text{If } (f, s) \in A_{train} \\ 0 & \text{Otherwise} \end{cases} . \quad (1)$$

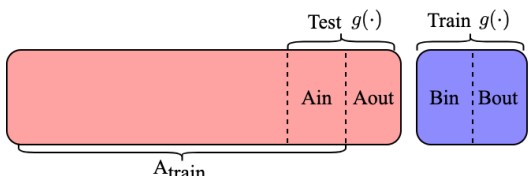

Figure 1: Data Splitting.

## 3.2 Shadow Models and Data Splitting

In this work, we follow the typical settings in the MI attack (Shokri et al., 2017; Hisamoto et al., 2020; Jagannatha et al., 2021; Shejwalkar et al., 2021) and assume Bob has access to the data from the same distribution as Alice to train their shadow models. Subsequently, Bob utilizes the known data membership of the shadow models as training labels to train an attack classifier $g(\cdot)$, whose goal is to predict the data membership of the shadow model. If the attack on the shadow model proves successful, Bob can employ the trained attack classifier to attempt an attack on Alice's model.

As depicted in Figure 1, we follow the previous work setting (Hisamoto et al., 2020) and split the whole dataset as $A_{all}$ and $B_{all}$, with Alice only having access to $A_{all}$ and Bob only has access to $B_{all}$. For Alice, $A_{all}$ is further split into two parts: $A_{train}$ and $A_{out}$, where $A_{train}$ is utilized for training a summarization model and $A_{out}$ serves as a hold-out dataset that is not used. (Note that $A_{train}$ includes the data used for validation and testing, and we use $A_{train}$ to specify the data used to train the model). In the case of Bob, $B_{all}$ is further split into $B_{in}$ and $B_{out}$, where Bob employs $B_{in}$ to train shadow models and $B_{out}$ serves as a hold-out dataset. To construct the attack classifier $g(\cdot)$, Bob can train $g(\cdot)$ with the objective of differentiating samples in $B_{in}$ and $B_{out}$.

## 3.3 Evaluation Protocols

We adopt the following evaluation protocols to evaluate $g(\cdot)$ attack performance on Alice's model. Given a document and its corresponding reference summary $(f, s)$, selected from $A_{train}$ or $A_{out}$, Bob sends $f$ to Alice's API and gets the output summary $\hat{s}$. Then Bob employs the trained classifier $g(\cdot)$ to infer whether the pair $(f, s)$ is present in Alice's training data. Since $A_{train}$ is much larger than $A_{out}$, we make the binary classification task more balanced by sampling a subset $A_{in}$ from the $A_{train}$ with the same size as $A_{out}$. Given a set $F$ of test samples $(f, s, \hat{s}, m)$, where $(f, s) \in A_{in} \cup A_{out}$,

$m$ is the ground truth membership, the Attack Accuracy (ACC) is defined as:

$$\text{ACC}(g, F) = \frac{1}{|F|} \sum^{F} [g(f, s, \hat{s}) = m], \quad (2)$$

where an accuracy above 50% can be interpreted as a potential compromise of privacy. Following a similar definition, we can define other commonly used metrics, such as Recall, Precision, AUC, etc.

Previous literature mainly uses accuracy or AUC to evaluate the privacy risk (Song and Shmatikov, 2019; Hisamoto et al., 2020; Mahloujifar et al., 2021; Jagannatha et al., 2021). However, these metrics only consider an average case and are not enough for security analysis (Carlini et al., 2022). Consider comparing two attackers: Bob$_1$ perfectly infers membership of 1% of the dataset but succeeds with a random 50% chance on the rest. Bob$_2$ succeeds with 50.5% on all dataset. On average, two attackers have the same attack accuracy or AUC. However, Bob$_1$ demonstrates exceptional potency, while Bob$_2$ is practically ineffective. In order to know if Bob can reliably infer the membership in the dataset (even just a few documents), we need to consider the low False-Positive Rate regime (FPR), and report an attack model's True-Positive Rate (TPR) at a low false-positive rate. In this work, we adopt the metric TPR$_{0.1\%}$, which is the TPR when FPR = 0.1%.

## 4 MI Attacks for Summarization Tasks

### 4.1 A Naive Baseline

The baseline attack is based on the observation that the generated summaries of training data often exhibit higher similarity to the reference summary, i.e., lower loss value (Varis and Bojar, 2021). In an extreme case, the model memorizes all training document-reference summary pairs and thus can generate perfect summaries for training samples ($s = \hat{s}$). Hence, it is natural for Bob to exploit the similarity between $\hat{s}$ and $s$ as a signal for membership inference.

There are multiple approaches to quantifying text similarity. Firstly, Bob utilizes the human-design metrics ROUGE-1, ROUGE-2, and ROUGE-L scores to calculate how many semantic content units from reference texts are covered by the generated summaries (Lin, 2004; Lin and Och, 2004). Additionally, Bob can adopt neural-based language quality scores, e.g., sentence transformer score(Reimers and Gurevych, 2019), to capture the

semantic textual similarity. Finally, we follow studies in the computer vision domain and also leverage the confidence score, such as the perplexity score, as the MI attack feature. Bob then concatenates all features to one vector, i.e., [ROUGE-1, ROUGE-2, ROUGE-L, Transformer Score, Confidence Score], and employs classifiers, such as random forest and multi-layer perceptron, to differentiate training and non-training samples. The baseline attack can be written as follows:

$$g(\cdot)_{Base} = g(\text{sim}(\hat{s}, s)), \qquad (3)$$

where sim represents the function that takes two summaries as input and returns a vector of selected similarity evaluation scores.

## 4.2 Document Augmentation for MI

The baseline attack is limited to relying solely on text similarity information from a single query. However, if Bob has the capability to send multiple queries to Alice's API, can Bob potentially explore more nuanced membership signals? Building upon this concept, we propose utilizing the robustness of output summaries when subjected to perturbations in input documents as the attack feature. Following a max-margin perspective, samples that exhibit high robustness are training data points (Tanay and Griffin, 2016; Hu et al., 2019; Deniz et al., 2020; Choquette-Choo et al., 2021). For our task, the assumption is that documents in the training dataset are more robust to perturbations and will have less change in the output summaries. Bob can consider various augmentation methods, such as word synonym replacement and sentence swapping. We use $D$ to denote the set of augmentation methods. Given a document $f$, Bob chooses one augmentation method $d \in D$ and generates $n$ new documents, i.e., $f_1^d, ... f_n^d$. Bob then queries the API using the augmented documents and obtains the output summaries $(f, f_1^d, ..., f_n^d) \rightarrow (\hat{s}, \hat{s}_1^d, ..., \hat{s}_n^d)$. To train the classifier, Bob uses similarity scores between all summaries with the reference summary as the feature, which can be written as follows:

$$g(\cdot)_{Aug} = g([\text{sim}(\hat{s}, s), \text{sim}(\hat{s}_1^d, s), ..., \text{sim}(\hat{s}_n^d, s)]). \qquad (4)$$

Compared to eq. 3, the proposed $g(\cdot)_{Aug}$ can additionally use the summaries' robustness information for MI, e.g., the variance of similarity scores $var((\hat{s}, s), ..., \text{sim}(\hat{s}_1^d, s))$.

### 4.2.1 Document-only MI Attack

Existing attack methods need Bob to access both the document $f$ and its corresponding reference summary $s$ to perform membership inference. However, it is challenging for Bob to obtain both of these for summarization tasks. Here, we propose a low-resource attack scenario: Bob only has a document and aims to determine whether the document is used to train the model. Under this scenario, the previously proposed attacks cannot be applied as there is no reference summary available.

However, the concept of evaluating sample robustness offers a potential solution as we can approximate the robustness without relying on reference summaries. To address this, we modify the $g(\cdot)_{aug}$: Instead of calculating the similarity scores between generated summaries $(\hat{s}, \hat{s}_1^d, ..., \hat{s}_n^d)$ and reference summaries $s$, Bob replaces the reference summary $s$ with the generated summary $\hat{s}$, and estimate the document robustness by calculating the similarity scores between $\hat{s}$ and perturbed documents' summaries $(\hat{s}_1^d, ..., \hat{s}_n^d)$. The proposed document only MI attack can be written as follows:

$$g(\cdot)_{D\_only} = g([\text{sim}(\hat{s}_1^d, \hat{s}), ..., \text{sim}(\hat{s}_n^d, \hat{s})])). \qquad (5)$$

Compared to $g(\cdot)_{aug}$, the proposed $g(\cdot)_{D\_only}$ obtains robustness information for the document only with the generated summary $\hat{s}$. Our experiments show that this approximate robustness contains valuable membership signals, and $g(\cdot)_{Donly}$ can effectively infer the membership of specific samples using only the documents as input.

## 5 Experiment Setup

### 5.1 Dataset

We perform our summarization experiments on three datasets: SAMsum, CNN/DailyMail (CNNDM), and MIMIC-cxr (MIMIC).

**SAMsum** (Gliwa et al., 2019) is a dialogue summarization dataset, which is created by asking linguists to create messenger-like conversations. Another group of linguists annotates the reference summary. The original split includes 14,732/818/819 dialogue-summary pairs for training/validation/test.

**CNNDM** (Hermann et al., 2015) is a news article summarization dataset. The dataset collects news articles from CNN and DailyMail. The summaries are created by human annotators. The original split includes 287,227/13,368/11,490 news article-summary pairs for training/validation/test.

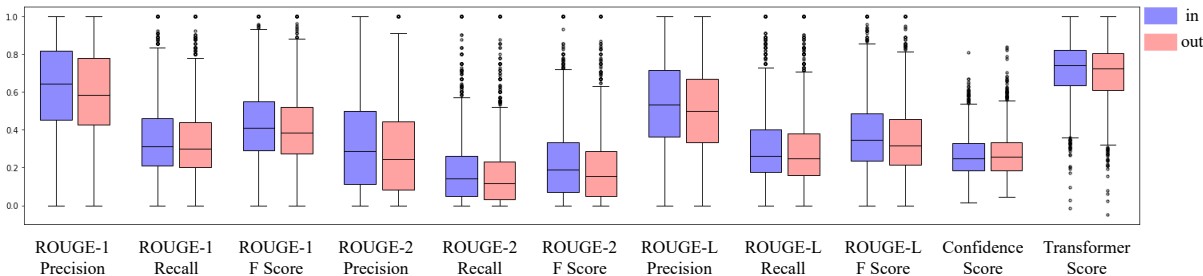

Figure 2: Distribution of similarity scores of $A_{in}$ and $A_{out}$ of SAMsum dataset.

| | SAMsum | | | CNNDM | | | MIMIC | | |
|---|---|---|---|---|---|---|---|---|---|
| | ACC | AUC | $\text{TPR}_{0.1\%}$ | ACC | AUC | $\text{TPR}_{0.1\%}$ | ACC | AUC | $\text{TPR}_{0.1\%}$ |
| RF | 61.10 | 64.72 | 1.31 | **53.48** | 55.38 | 0.83 | 65.41 | 66.37 | 2.58 |
| LR | 61.15 | **65.88** | 1.05 | 51.24 | 53.88 | 0.05 | 66.73 | 68.64 | 2.20 |
| SVM | 61.67 | 65.45 | 2.03 | 50.30 | 52.72 | 0.03 | 65.90 | 69.24 | 2.34 |
| MLP | **61.73** | 65.84 | **2.15** | 52.33 | **55.84** | **1.17** | **67.11** | **70.71** | **3.05** |
| RoBERTa | 60.10 | 63.27 | 1.26 | 50.01 | 51.71 | 0.05 | 66.08 | 68.01 | 2.05 |

Table 1: Baseline Attack Results. Bob tried different classifiers, including Random Forest (RF), Logistic Regression (LR), Support Vector Machine (SVM), and Multi-layer Perceptron (MLP). Following the evaluation protocol in Section 3.3, we show membership attack performance on $A_{in}$ and $A_{out}$.

**MIMIC** is a public radiology report summarization dataset. We adopt task 3 in MEDIQA 2021 (Abacha et al., 2021), which aims to generate the impression section based on the findings and background sections of the radiology report. We choose the MIMIC-cxr as the data source, and the original split includes 91544/2000 medical report-impression pairs for training/validation.

As we discussed in Sec. 3.2, we reorganized the datasets into three disjoint sets: $A_{train}$, $A_{out}$, and $B_{all}$. We assume Bob can access around 20% of the dataset to train shadow models and $g(\cdot)$. In Table 2, we show the details number for each split.

| | $A_{train}$ | $A_{in}$ | $A_{out}$ | $B_{all}$ |
|---|---|---|---|---|
| SAMsum | 13,369 | 1,000 | 1,000 | 2,000 |
| CNNDM | 252,085 | 20,000 | 20,000 | 40,000 |
| MIMIC | 78,544 | 10,000 | 10,000 | 20,000 |

Table 2: Each dataset is divided in to three disjoint sets: $A_{train}$, $A_{out}$ and $B_{all}$. $A_{in}$ is sampled from $A_{train}$ with a same size as $A_{out}$.

## 5.2 Models and Training Details

In our experiments, we adopted two widely used summarization models: BART-base (Lewis et al., 2020) and FLAN-T5 base (Chung et al., 2022) (Results of FLAN-T5 are detailed in the Appendix).

We adopt Adam (Kingma and Ba, 2014) as the optimizer. For SAMsum, CNNDM, and MIMIC, the batch size is set as 10/4/4, and the learning rate is set as $2e^{-5}$, $2e^{-5}$, $1e^{-5}$. During inference, we set the length penalty as 2.0, the beam search width as 5, and the max/min generation length as 60/10, 140/30, and 50/10. Alice chooses the best model based on the validation ROUGE-L score. Bob randomly splits $B_{all}$ into two equal parts: $B_{in}$ and $B_{out}$. Bob employs $B_{in}$ to train shadow models and chooses the best model based on the validation ROUGE-L performance. We only trained one shadow model in the experiment. Our implementation is based on the open-source PyTorch-transformer repository. [1] All experiments are repeated 5 times and report the average results.

## 5.3 Augmentation Methods

We consider three augmentation methods: word synonym (WS), sentence swapping (SW), and back translation (BT)[2]: word synonyms randomly choose 10% of words in a document and change to their synonym from WordNet(Miller, 1995), sentence swapping randomly chooses a sentence and swap it with another sentence, back translation first translates the document to French, Spanish, and

---

[1]https://github.com/huggingface/transformers

[2]Our implementation is based on Data Augmentation by Back-translation (DAB). Github: https://github.com/vietai/dab

| | | SAMsum | | | CNNDM | | | MIMIC | | |
|---|---|---|---|---|---|---|---|---|---|---|
| | | ACC | AUC | TPR$_{0.1\%}$ | ACC | AUC | TPR$_{0.1\%}$ | ACC | AUC | TPR$_{0.1\%}$ |
| RF | Base | 61.10 | 64.72 | 1.31 | 53.48 | 55.38 | 0.83 | 65.41 | 66.37 | 2.58 |
| | WS | 62.11 | 65.23 | 1.45 | **54.69** | 56.21 | 0.95 | 65.53 | 66.53 | 2.61 |
| | SW | **62.51** | **65.81** | **1.61** | 54.51 | **56.83** | **1.01** | **66.44** | **66.48** | **2.65** |
| | BT | 61.07 | 64.41 | 1.31 | 53.30 | 54.96 | 0.78 | 65.22 | 65.98 | 2.49 |
| LR | Base | 61.15 | 65.88 | 1.05 | 51.24 | 53.88 | 0.05 | 66.73 | 68.64 | 2.20 |
| | WS | 61.23 | 65.90 | 1.05 | 52.25 | 53.94 | 0.13 | 67.01 | 68.74 | 2.25 |
| | SW | **62.03** | **66.70** | **1.20** | **52.58** | **54.92** | **0.21** | **67.59** | **69.15** | **2.86** |
| | BT | 60.14 | 65.13 | 1.01 | 52.13 | 53.95 | 0.06 | 66.78 | 68.99 | 2.21 |
| SVM | Base | 61.67 | 65.45 | 2.03 | 50.30 | 52.72 | 0.03 | 65.90 | 69.24 | 2.34 |
| | WS | 62.26 | 66.03 | 2.20 | 50.45 | 52.87 | 0.11 | 66.93 | 70.41 | 2.43 |
| | SW | **62.75** | **66.55** | **2.41** | **51.67** | **53.89** | **0.13** | **66.87** | **70.59** | **2.52** |
| | BT | 61.70 | 65.51 | 2.05 | 50.41 | 52.81 | 0.05 | 65.85 | 68.94 | 2.27 |
| MLP | Base | 61.73 | 65.84 | 2.15 | 52.33 | 55.84 | 1.17 | 67.11 | 70.71 | 3.05 |
| | WS | 62.13 | 66.21 | **2.51** | 53.11 | 56.21 | 1.25 | **68.24** | 71.12 | 3.15 |
| | SW | **62.85** | **67.00** | 2.49 | **53.53** | **56.26** | **1.36** | 68.18 | **71.33** | **3.57** |
| | BT | 62.01 | 66.81 | 2.17 | 52.40 | 55.91 | 1.19 | 67.01 | 70.69 | 3.02 |

Table 3: Document Augmentation Attack Results. *Base* shows the baseline attack results in Table 1.

| | SAMsum | | | CNNDM | | | MIMIC | | |
|---|---|---|---|---|---|---|---|---|---|
| | ACC | AUC | TPR$_{0.1\%}$ | ACC | AUC | TPR$_{0.1\%}$ | ACC | AUC | TPR$_{0.1\%}$ |
| RF | 57.11 | **58.24** | 1.27 | 51.07 | 53.09 | 0.52 | **60.17** | **61.25** | 2.13 |
| LR | 57.03 | 57.85 | 1.10 | 50.89 | 52.83 | 0.11 | 57.72 | 59.44 | 2.13 |
| SVM | 57.05 | 57.11 | 1.89 | 50.41 | 52.63 | 0.09 | 59.15 | 61.22 | 1.91 |
| MLP | **57.21** | 57.05 | **1.97** | **51.30** | **53.11** | **1.07** | 60.07 | 60.21 | **2.67** |

Table 4: Document only Attack Results based on sentence swapping augmentation.

German, and then back translates to English. In our main experiments, we generate 6 augmented samples for WS, SW, and 3 for BT.

# 6 Experiment Results

**Baseline Attack.** We present the results of the baseline attack in Table 1. Our analysis reveals that the attack is successful in predicting membership, as the accuracy and AUC results on the three datasets are above 50%. Furthermore, the attack AUC on the MIMIC and SAMsum datasets is above 65%, which highlights a significant privacy risk to Alice's model. In Figure 2, we examine the feature distribution of $A_{in}$ and $A_{out}$. Our key observation is that $A_{in}$ exhibits notably higher ROUGE-1, ROUGE-2, and Transformer Scores than $A_{out}$, indicating that the model's behavior is distinct on training and non-training samples. Additionally, our study discovered that the confidence score, which has been found to be useful in pre-

vious classification models (Shokri et al., 2017), is useless for the summarization model. Furthermore, we fine-tune a RoBERTa model and use the raw texts to differentiate generated summaries of $B_{in}$ and $B_{out}$, referred to as RoBERTa in the table. However, the results indicate that raw text is inferior to the similarity score features, with the MLP model using similarity scores as features achieving the best performance.

In addition to the AUC and ACC scores, we also evaluate the performance of the attacks in the high-confidence regime. Specifically, we report the true positive rate under a low false positive rate of 0.1%, referred to as TPR$_{0.1\%}$ in the table. Our results demonstrate that the model can reliably identify samples with high confidence. For example, the MLP model achieves a TPR$_{FPR}0.1\%$ of 3.05% on the MIMIC dataset, which means that the model successfully detects 305 samples in $A_{in}$ with only 10 false positives in $A_{out}$.

**Document Augmentation MI Attack.** In this section, we investigate the effectiveness of evaluating the model's robustness against document modifications as the MI feature. The results of the attack are presented in Table 3. We observe a consistent improvement in attack performance across all datasets compared to $g(\cdot)_{Base}$. Specifically, the improvement in $\text{TPR}_{0.1\%}$ indicates that the robust signal allows the attacker to detect more samples with high confidence. We find that sentence swapping achieves the best attack performance across all datasets. In Figure, we show the standard deviation distribution of ROUGE-L F1 scores, 3, calculated as $\text{SD}(\text{R-L}(\hat{s}, s), \text{R-L}(\hat{s}_1^d, s), ..., \text{R-L}(\hat{s}_n^d, s))$. We find that the variance of training data is notably lower than non-training data, indicating that training samples are more robust against perturbations.

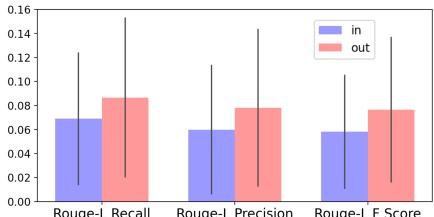

Figure 3: ROUGE-L Standard deviation of $g(\cdot)_{D\_aug}$.

**Document-only Attack.** In this section, we present the results of our document-only attack. As previously discussed in Section 4.2.1, the attack classifier $g(\cdot)_{D\_only}$ does not have access to reference summaries. Instead, Bob estimates the model's robustness by using generated summaries. In Figure 4, we show the standard deviation distribution of ROUGE-L scores, calculated as $\text{SD}(\text{R-L}(\hat{s}_1^d, \hat{s}), \text{R-L}(\hat{s}_2^d, \hat{s}), ..., \text{R-L}(\hat{s}_n^d, \hat{s}))$. Similar to the results in Figure 3, the variance of training data is lower than that of non-training data but with smaller differences. Reflecting on the results, we observe a lower attack performance for document-only attacks (Table 4) compared to $g(\cdot)_{Base}$ in Table 1. However, attack accuracy and AUC are above 50%, indicating a privacy risk even under this low-resource attack. More importantly, the $\text{TPR}_{0.1\%}$ results show that Bob can still infer certain samples' membership with high confidence.

## 7 Ablation Studies

In this section, we will investigate several impact factors in MI attacks. All experiments were conducted using the baseline attack with the MLP classifier. A more detailed analysis is in the Appendix.

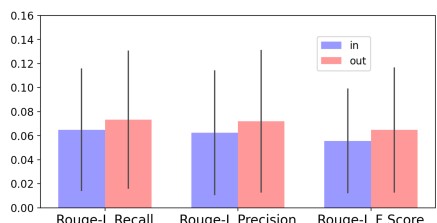

Figure 4: ROUGE-L Standard deviation of $g(\cdot)_{D\_only}$.

**Impact of Overfitting.** In Figure 5, we show the attack AUC and validation ROUGE-L F1 score under varying training steps on the SAMsum dataset. We find that the attack AUC increases steadily as the number of Alice's training steps increases, which is consistent with previous research (Shokri et al., 2017). Moreover, early stopping by ROUGE score (5 epochs) cannot alleviate the attack. The AUC curve indicates that the model gets a high attack AUC at this checkpoint. A better early stop point is 3 epochs, which significantly reduces the MI attack AUC without a substantial performance drop. However, in practice, it is hard to select a proper point without relying on an attack model.

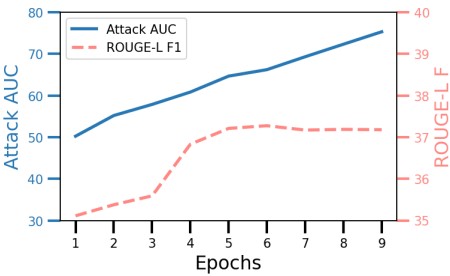

Figure 5: Attack AUC under different epochs.

**Impact of Dataset Size.** In this study, we assess the impact of dataset size on MI attacks. To do this, we train our model with 10% to 100% of the total dataset. Our results, as depicted in Figure 6, indicates that as the size of the training set increases, the AUC of MI attacks decreases monotonically for both SAMsum and CNNDM dataset. This suggests that increasing the number of samples in the training set can help to alleviate overfitting and reduce the MI attack AUC. Some recent studies have highlighted the issue of duplicate training samples in large datasets (Lee et al., 2022). This duplication can escalate the privacy risks associated with these samples and should be taken into consideration when employing large datasets.

**Impact of the Model Architecture.** In previous experiments, we assumed that Bob uses the same

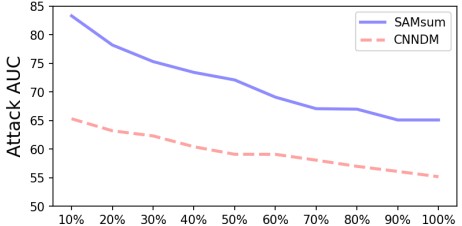

Figure 6: Attack AUC across different dataset sizes.

architecture as Alice to train shadow models. In this section, we further explore the attack transferability attack across different model architectures. As shown in Figure 7, Bob and Alice can choose different model architectures, we evaluate the transferability metrics for various models on SAMsum Dataset, including BART, BertAbs (Liu and Lapata, 2019), PEGASUS (Zhang et al., 2020a), and FLAN-T5 (Chung et al., 2022). The results indicate that the attack AUC is highest when both Bob and Alice employ the same model. However, even when Bob and Alice utilize different models, the MI attack exhibits considerable transferability across the selected model architectures. These findings suggest that the membership signal exploited by the attack classifier demonstrates generalizability and effectiveness across various architectures.

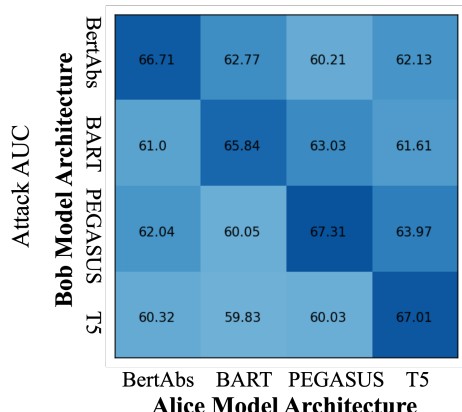

Figure 7: Architecture Transferability.

## 8   Defense Methods

We now investigate some approaches that aim to limit the model's ability to memorize its training data. Specifically, we try two approaches: differential privacy SGD (DP-SGD) [3] (Dwork, 2008; Machanavajjhala et al., 2017; Li et al., 2021) and

---

[3] Our implementation is based on dp-transformers. Github: https://github.com/microsoft/dp-transformers

$L_2$ regularization (Song et al., 2019). For DP-SGD, $\epsilon$ is the privacy budget, where a lower $\epsilon$ indicates higher privacy. We conduct experiments on the SAMsum dataset. As shown in Table 5, we find that as the $\lambda$ increase and $\epsilon$ decrease, the attack AUC stably drops. Particularly, when $\epsilon = 8.0$ and $\lambda = 12.0$, the AUC drops to about 50%. However, we find defense methods cause a notable performance drop on the ROUGE-L F1 score. Indicating that there is a privacy-utility trade-off.

| DP-SGD | | | |
|---|---|---|---|
| $\epsilon$ | 200.0 | 100.0 | 8.0 |
| AUC | 64.12 | 54.51 | 50.46 |
| ROUGE-L | 37.21 | 32.32 | 27.31 |
| | | | |
| L2 Regularization | | | |
| $\lambda$ | 0.0 | 6.0 | 12.0 |
| AUC | 65.84 | 59.34 | 52.64 |
| ROUGE-L | 37.35 | 34.32 | 29.11 |

Table 5: Defense Performance on DP-SGD and L2 Regularization with different privacy strengths.

## 9   Limitations

In this work, we demonstrate that the MI attack is effective. However, it remains unclear what properties make samples more susceptible to MI attack. In other words, given a model and a dataset, we cannot predict which samples are more likely to be memorized by the model. We find that the detected samples under $TPR_{0.1\%}$ have an average shorter reference length, but further research is needed to fully answer this question. Additionally, it is important to note that while the MI attack is a commonly used attack, its privacy leakage is limited. Other attacks pose a more significant threat in terms of information leakage (Carlini et al., 2021). The evaluation of these attacks in summarization tasks should be prioritized in future studies.

## 10   Conclusion

In this paper, we investigated the membership inference attack for the summarization task and explored two attack features: text similarity and data robustness. Experiments show that both features contain fine-grained MI signals. These results reveal the potential privacy risk for the summarization model. In the future, we would like to explore advanced defense methods and alleviate the trade-off between privacy and utility.

## Acknowledgement

The authors thank the anonymous reviewers for their helpful comments. The work is in part supported by NSF grants NSF CNS-1816497, IIS-1849085, and IIS-2224843. The views and conclusions contained in this paper are those of the authors and should not be interpreted as representing any funding agencies.

## Ethics Statement

In this study, we have utilized publicly available datasets and models to ensure the transparency and reproducibility of our research. We ensure that these datasets do not disclose any private, protected information relating to individuals. We wish to clarify that the content encompassed in these datasets does not reflect our personal views or opinions. Our analysis and findings are solely based on the data provided and aim to enhance understanding of language generation models and their security implications.

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

## A   Experiments on More Models

In our primary experiments, our focus is on the BART model; however, we are also interested in exploring other commonly used models. Among these models, we consider the Fan-T5 base (Chung et al., 2022), where we maintain the same experimental setup but replace the Alice and Bob model with Flan-T5. Table 7 presents the baseline attack results, which are consistent with our findings in the BART model. Our analysis demonstrates the successful prediction of membership to the Flan-T5 model, as evidenced by the accuracy and AUC results across all three datasets exceeding 50%. Furthermore, the attack AUC for the MIMIC and SAMsum datasets exceeds 66%, highlighting a significant privacy risk to Alice's model.

Additionally, Table 8 showcases the results of the Document augmentation attack, revealing a consistent improvement in attack performance across all datasets compared to $g(\cdot)_{Base}$. Notably, the enhancement in $\text{TPR}_{0.1\%}$ indicates that the robust signal enables the attacker to detect more samples with high confidence. Our findings indicate that sentence swapping yields the most effective attack performance across all datasets and metrics.

Moreover, in Table 9, we observe a lower attack performance for document-only attacks compared to $g(\cdot)_{Base}$ in Table 7. Nevertheless, the attack accuracy and AUC remain above 50%, signifying a privacy risk even in the context of this low-resource attack. Most importantly, the $\text{TPR}_{0.1\%}$ results demonstrate that Bob can still infer the membership of certain samples with high confidence.

To conclude, our findings are consistent across both the Flan-T5 and BART models, indicating that these summarization models have the ability to memorize training data and pose a valid threat of leaking membership information.

## B   Feature Importance

In this section, we delve into the feature importance of the baseline MI attack, specifically targeting the SAMsum dataset. Figure 8 and 9 displays the feature importance scores[4] as determined by the Random Forest classifier for the baseline attack. The ROUGE-2 F1 score emerges as the most valuable feature. On the contrary, the confidence score,

---

[4]The importance score is based on the scikit-learn package: https://scikit-learn.org/stable/auto_examples/ensemble/plot_forest_importances.html

---

despite its crucial role in MI attacks within the computer vision domain, proves to be insignificant in the sequence-to-sequence model. This could be attributed to the beam search process, which invariably samples sentences with high confidence, thereby rendering this feature redundant.

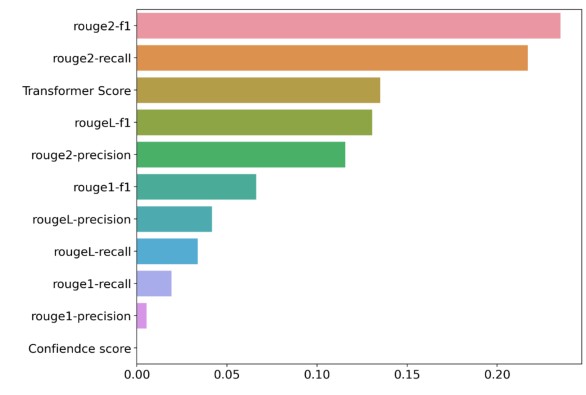

Figure 8: Feature Importance in MI attack on BART model.

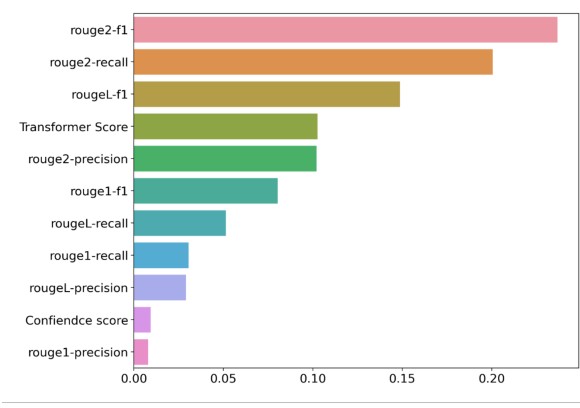

Figure 9: Feature Importance in MI attack on FLAN-T5 model.

## C   More on Ablation Studies

In this section, we will add more ablation studies. Firstly, we will study the impact of overfitting, and dataset size on the FLAN-T5 dataset. Then we will introduce the impact of query numbers. All experiments were conducted with the baseline attack method, employing the MLP classifier.

**Impact of Overfitting.** We study the impact of overfitting in MI attacks on the FLAN-T5 model. Figure 10 shows the attack AUC and validation ROUGE-L F1 score under varying training steps on the SAMsum dataset. We observe that the attack AUC increases steadily as the number of Alice's training steps increases. Early stopping by ROUGE

score (4 epochs) cannot alleviate the attack.

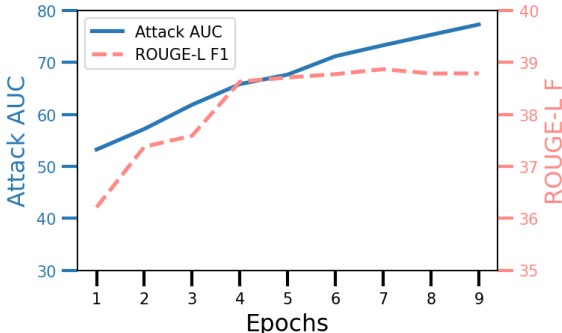

Figure 10: Attack AUC under different epoches.

**Impact of Dataset Size.** In this study, we assess the impact of dataset size on MI attacks on the FLAN-T5 model. Specifically, we train the model with 10% to 100% of the total dataset. As depicted in Figure 11, we found that as the size of the training set increases, the AUC of MI attacks decreases monotonically for both SAMsum and CNNDM datasets. This outcome aligns with our observations from the BART model and suggests that increasing the number of samples in the training set can help to alleviate overfitting.

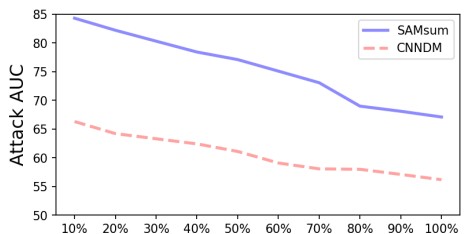

Figure 11: Attack AUC across various dataset sizes.

**Impact of Augmentation Numbers.** In our primary experiments involving document augmentation for MI attacks, we generated 6 augmentations for word synonym (WS) and sentence swapping (SW), and 3 for back translation (BT). In this section, we extend our investigation to the impact of augmentation quantity, focusing on WS and SW, as they exhibit a significant increase over the baseline attack. Table 6 presents the attack AUC for augmentations of 6, 12, and 24 on the SAMsum dataset. Notably, we observe a slight improvement in attack performance with an increased number of augmentation samples, which aligns with the notion that more augmented data can help the classifier better evaluate the sample's robustness. However, it's important to acknowledge that a higher augmentation

number necessitates more queries to the API, consequently escalating the attack cost. Additionally, we attempted to combine data from the two augmentation methods, with the results documented in the 'Comb' row. Interestingly, merging data from the two augmentation methods did not further enhance the attack performance.

| BART | | | |
|---|---|---|---|
| | 6 | 12 | 24 |
| WS | 66.21 | 66.34 | 66.39 |
| SW | 67.00 | 67.14 | 67.31 |
| Comb | 66.65 | 66.94 | 67.13 |

| FLAN-T5 | | | |
|---|---|---|---|
| | 6 | 12 | 24 |
| WS | 67.01 | 67.19 | 67.35 |
| SW | 68.22 | 68.29 | 68.37 |
| Comb | 67.27 | 67.33 | 67.84 |

Table 6: Different Augmentation Number.

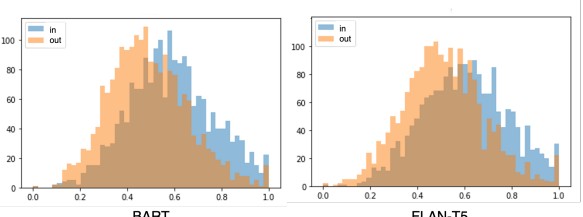

Figure 12: Architecture Transferability.

## D  More on Architecture Transferability

As discussed in Section 7, when Alice and Bob employ different architectures, the MI attack exhibits robust architecture transferability. This implies that even when the attacker's classifier is trained on a shadow model distinct from Alice's model, the learned attack signal is generalizable across different architectures. In this section, we aim to explain this transferability. Our key observation is that the MI signal remains remarkably consistent, even across different architectures. As demonstrated in Figure 12, we exhibit the distribution of the ROUGE-L F1 score in Alice's model, using both the BART and FLAN-T5 architectures. The observation reveals a consistent pattern: the ROUGE score for the training data is notably higher than that for non-training data. This trend persists across both architectures, which lends insight into the attack's transferability.

|  | SAMsum | | | CNNDM | | | MIMIC | | |
|---|---|---|---|---|---|---|---|---|---|
|  | ACC | AUC | $TPR_{0.1\%}$ | ACC | AUC | $TPR_{0.1\%}$ | ACC | AUC | $TPR_{0.1\%}$ |
| RF | 61.87 | 65.01 | 1.22 | 53.01 | 56.99 | 0.81 | 66.23 | 67.11 | 2.63 |
| LR | 62.43 | 66.15 | 1.11 | 52.13 | 55.01 | 0.11 | 67.32 | 69.51 | 2.30 |
| SVM | 63.14 | 67.00 | 1.95 | 52.21 | 54.41 | 0.06 | 66.41 | 68.71 | 2.13 |
| MLP | 63.10 | 67.01 | 2.13 | 53.14 | 56.32 | 1.17 | 67.91 | 72.05 | 3.18 |
| RoBERTa | 60.45 | 64.33 | 1.27 | 50.48 | 53.28 | 0.75 | 65.71 | 69.83 | 2.37 |

Table 7: FLAN-T5 Baseline Attack Results. Following the evaluation protocol in Section 3.3, we show $g(\cdot)_{Base}$ performance on $A_{in}$ and $A_{out}$.

|  |  | SAMsum | | | CNNDM | | | MIMIC | | |
|---|---|---|---|---|---|---|---|---|---|---|
|  |  | ACC | AUC | $TPR_{0.1\%}$ | ACC | AUC | $TPR_{0.1\%}$ | ACC | AUC | $TPR_{0.1\%}$ |
| RF | Base | 61.87 | 65.01 | 1.22 | 53.01 | 56.99 | 0.81 | 66.23 | 67.11 | 2.63 |
|  | WS | 62.43 | 66.55 | 1.24 | 55.21 | 56.52 | 0.77 | 68.03 | 68.03 | 2.71 |
|  | SW | 63.00 | 66.84 | 1.66 | 55.11 | 56.25 | 1.13 | 67.91 | 68.17 | 3.31 |
|  | BT | 61.22 | 65.18 | 1.20 | 55.13 | 56.77 | 0.95 | 66.37 | 67.00 | 2.43 |
| LR | Base | 62.43 | 66.15 | 1.11 | 52.13 | 55.01 | 0.11 | 67.32 | 69.51 | 2.30 |
|  | WS | 61.53 | 66.21 | 1.18 | 53.45 | 55.31 | 0.20 | 68.13 | 69.07 | 2.43 |
|  | SW | 63.35 | 67.99 | 1.28 | 54.00 | 55.27 | 0.33 | 68.55 | 71.35 | 2.83 |
|  | BT | 61.43 | 66.01 | 1.13 | 53.01 | 54.22 | 0.19 | 68.13 | 70.59 | 3.05 |
| SVM | Base | 63.14 | 67.00 | 1.95 | 52.21 | 54.41 | 0.06 | 66.41 | 68.71 | 2.13 |
|  | WS | 63.54 | 66.35 | 2.24 | 52.77 | 55.45 | 0.19 | 67.10 | 69.58 | 2.33 |
|  | SW | 63.17 | 67.12 | 2.12 | 52.59 | 55.32 | 0.27 | 68.15 | 70.83 | 3.01 |
|  | BT | 62.15 | 67.08 | 2.14 | 50.75 | 54.55 | 0.13 | 65.85 | 68.11 | 1.99 |
| MLP | Base | 63.10 | 67.01 | 2.13 | 53.14 | 56.32 | 1.17 | 67.91 | 72.05 | 3.18 |
|  | WS | 62.22 | 67.01 | 2.77 | 54.34 | 57.12 | 1.21 | 68.56 | 71.22 | 4.01 |
|  | SW | 63.15 | 68.22 | 3.71 | 54.99 | 58.18 | 2.04 | 68.33 | 73.55 | 3.88 |
|  | BT | 62.22 | 67.21 | 2.20 | 53.74 | 57.73 | 1.56 | 67.13 | 72.14 | 3.40 |

Table 8: Document Augmentation Attack Results on FLAN-T5. $Base$ shows the baseline attack results in Table 6.

|  | SAMsum | | | CNNDM | | | MIMIC | | |
|---|---|---|---|---|---|---|---|---|---|
|  | ACC | AUC | $TPR_{0.1\%}$ | ACC | AUC | $TPR_{0.1\%}$ | ACC | AUC | $TPR_{0.1\%}$ |
| RF | 56.30 | 57.74 | 1.17 | 50.99 | 54.08 | 0.57 | 60.55 | 61.65 | 2.22 |
| LR | 57.31 | 59.88 | 1.05 | 51.22 | 53.76 | 0.39 | 55.01 | 58.83 | 2.26 |
| SVM | 57.15 | 58.86 | 1.85 | 51.03 | 54.07 | 0.12 | 59.33 | 62.25 | 1.91 |
| MLP | 57.60 | 57.35 | 2.08 | 52.73 | 54.49 | 1.21 | 61.15 | 62.77 | 2.77 |

Table 9: Document only Attack Results based on sentence swapping augmentation on FLAN-T5. $Base$ shows the best baseline attack results in Table 6 with full knowledge.