# OpenReview forum: "Assessing Privacy Risks in Language Models: A Case Study on Summarization Tasks"
_EMNLP/2023/Conference — EMNLP 2023 Findings_

### Official Review · Reviewer_9rho · 2023-08-04

**Soundness:** 4

**Excitement:**

4: Strong: This paper deepens the understanding of some phenomenon or lowers the barriers to an existing research direction.

**Paper Topic And Main Contributions:**

This paper takes a close look at membership inference (MI) attacks for large language models, focusing on summarization tasks. It's well-written, easy to read, and the definitions and notations are clear. The figures in the paper are well-designed, adding to the understanding of the topic.



**Questions For The Authors:**

1) Is it possible to extend the authors' method to perform membership inference attacks on autoregressive generative language models? I'm intrigued to explore whether some publicly available models, displaying impressive capabilities, were trained on legally questionable data sets, such as non-permissive codebases.

2) Have the authors conducted experiments in this direction? If there is existing research pertaining to membership inference attacks on generative models, please consider adding it to the related work section.


**Reasons To Accept:**

The authors present a simple but effective method for MI attacks that works specifically with summarization. Unlike previous methods that focused on classification tasks, this approach doesn't need scores or probabilities and can work with just black-box API access. The method can even be used without the summarized document, making it very useful for real-world situations.

The paper also includes extra tests and analysis, such as how well the method can be transferred to different setups. The authors have thought about how to protect against these kinds of attacks and the balance between keeping data private and making it useful. Overall, this is a solid paper that adds valuable insights to the field of natural language processing and information security. It's a good read for anyone interested in these subjects.

**Reasons To Reject:**

At times the main inference was hard to follow and justify logical jumps, however, the abundance of figures/tables covered those minor discrepancies.

**Reproducibility:**

4: Could mostly reproduce the results, but there may be some variation because of sample variance or minor variations in their interpretation of the protocol or method.

**Reviewer Confidence:**

4: Quite sure. I tried to check the important points carefully. It's unlikely, though conceivable, that I missed something that should affect my ratings.

---

> ### Author Rebuttal · Authors · 2023-08-29
>
> **Q1:** At times the main inference was hard to follow and justify logical jumps, however, the abundance of figures/tables covered those minor discrepancies.
>
> **R1:** Thank you for your feedback. We will enhance the narrative clarity and ensure it aligns well with the figures and tables for better understanding.
>
> **Q2:** Is it possible to extend the authors' method to perform membership inference attacks on autoregressive generative language models? I'm intrigued to explore whether some publicly available models, displaying impressive capabilities, were trained on legally questionable data sets, such as non-permissive codebases.
>
> **R2:** Thank you for your insightful suggestions.
> - Indeed, the proposed attack can be applied to autoregressive generative language models. Based on your suggestion, we tested our method on Vicuna-7B [1], which is recognized as one of the leading open-source autoregressive generative language models. We fine-tuned Vicuna on the Samsum and CNNDM datasets and applied the baseline and document augmentation attacks, utilizing MLP as the classifier. For augmentation, we selected the sentence swap method, which proved the most effective for membership inference attacks in our study. **As Table 1 depicts, Vicuna-7B is vulnerable to membership inference attacks,** evidenced by an attack AUC significantly greater than 50%. Additionally, the TPR_0.1\% results indicate that the attack can identify specific training data with high accuracy and confidence.
> - We agree with you that **membership inference attacks have the potential to help identify if specific protected content has been utilized in pretraining/fine-tuning generative models,** such as non-permissive codebases and copyrighted literature. Using membership inference attacks to safeguard intellectual property presents a compelling research opportunity, and we plan to delve deeper into it in our upcoming work.
>
>
> **Table 1: MI Attack Performance on Vicuna-7B Model**
> |MI Attack $\downarrow$| Dataset $\downarrow$|ACC|AUC|TPR_0.1\%|
> |:---:|:---:|:---:|:---:|:---:|
> |Baseline Attack| SAMsum | 61.87 | 65.43 | 1.23 |
> || CNNDM| 52.15 | 56.86| 0.29|
> |Document Augmentation Attack | SAMsum |62.17 | 65.79 |1.50 |
> ||CNNDM | 53.01 | 57.85 | 0.81 |
>
> **Q3:** Have the authors conducted experiments in this direction? If there is existing research pertaining to membership inference attacks on generative models, please consider adding it to the related work section.
>
> **R3:** Thank you for your insights.
> - To the best of our knowledge, **limited research exists specifically on membership inference attacks for text generative models.** We have referenced two prior studies that delve into this subject. However, these studies either require both data and its corresponding label for membership inference or demand white-box access to the model. Additionally, both of them focus on relatively small models, and membership inference attack on recent large generative models is under-studied.
> -  There are also studies that focus on membership inference attacks in different domains, such as image generation [2] and text-image generation [3]. We appreciate the suggestion and plan to include these references in our updated paper.
>
> **References**\
> [1] Chiang, Wei-Lin, et al. "Vicuna: An open-source chatbot impressing gpt-4 with 90%* chatgpt quality." 2023.\
> [2] Chen, Dingfan, et al. "Gan-leaks: A taxonomy of membership inference attacks against generative models." CCS 2020.\
> [3] Wu, Yixin, et al. "Membership inference attacks against text-to-image generation models." arxiv 2022.

---

### Official Review · Reviewer_NwCg · 2023-08-05

**Soundness:** 3

**Excitement:**

3: Ambivalent: It has merits (e.g., it reports state-of-the-art results, the idea is nice), but there are key weaknesses (e.g., it describes incremental work), and it can significantly benefit from another round of revision. However, I won't object to accepting it if my co-reviewers champion it.

**Paper Topic And Main Contributions:**

This paper tackles a concern that LLMs may disclose information in the training data. This paper focuses on the summarization task and investigate the membership inference (MI) attack. In this setting, this paper uses a sample and black-box access to a model to determine if the sample was part of the training data. This paper exploits text similarity between an output and the reference summary of its input text and the stability of model outputs to input text as modified with operations such as word synonym replacement and sentence swapping. Evaluation of their effectiveness on SAMsum, CNN and Daily Mail, and MIMIC  demonstrates that summarization models are at risk of exposing data membership, even in cases where the reference summary is not available. Furthermore, this paper also discusses safeguards for training summarization models to protect against MI attacks.



**Questions For The Authors:**

(A)	Are there any reasons that this paper focuses on summarization compared with previous work focusing on MT translation? The reviewer thinks this paper’s method can be applied to MT.

**Reasons To Accept:**

- Tackling a problem of a MI attack for summarization
- Proposal of input document only MI attack for summarization using input documents modified with word synonym replacement, sentence swapping and back translation.


**Reasons To Reject:**

-	The reason why this paper focuses on summarization is not presented. Previous work focused on Machine Translation (MT).

**Reproducibility:**

3: Could reproduce the results with some difficulty. The settings of parameters are underspecified or subjectively determined; the training/evaluation data are not widely available.

**Reviewer Confidence:**

4: Quite sure. I tried to check the important points carefully. It's unlikely, though conceivable, that I missed something that should affect my ratings.

**Typos Grammar Style And Presentation Improvements:**


-	The reviewer thinks description of more specific applications is appreciated by readers. What situation is this paper’s method used for? For example, Hisamoto et al., 2020 described them in their paper for the MT case.

---

> ### Author Rebuttal · Authors · 2023-08-29
>
> **Q1:** The reason why this paper focuses on summarization is not presented. Previous work focused on Machine Translation (MT).
>
> **R1:** Thank you for your insightful comments.
>
> - In the introduction section, we emphasized our focus on the summarization task for two primary reasons: **1) Summarization stands as a representative task for generative models**. This makes it a crucial domain to investigate, especially given the increasing adoption of such models in various applications. and **2) Given the widespread deployment of summarization models in critical sectors like healthcare**, there exists a potential risk of privacy breaches. Documents in such domains often contain Personally Identifiable Information (PII). Despite the potential for privacy risks, there is a lack of prior research specifically addressing these concerns in the context of summarization models.
> - The proposed membership inference attack methods and threat models are transferrable to many other sequence-to-sequence tasks, such as machine translation and generative question answering. However, adapting them requires the exploration and analysis of domain-specific MI features. We believe that the incorporation of additional domains within this study might overextend the scope of the paper, making it overly lengthy and potentially overwhelming readers. We plan to explore other domains in our future research.
>
> **Q2:** The reviewer thinks description of more specific applications is appreciated by readers. What situation is this paper’s method used for?
>
> **R2:** Thank you for the constructive suggestion.
> - As we emphasized in the paper, membership inference attacks are standard tests for privacy vulnerabilities. **From the attacker's perspective**, these attacks can identify whether a specific document is used for training the summarization model. **Conversely, for model developers,** the proposed attacks act as evaluative tools, allowing stakeholders to assess the potential privacy risks associated with their models.
> - As a concrete example, consider the hospital environment where summarization models are often employed to distill extensive patient histories and reports into concise summaries for efficient review by healthcare practitioners [1, 2]. However, there exists a tangible risk: **a summarization model trained on a comprehensive dataset of patient records might inadvertently remember and leak information about patients,** potentially violating regulations like HIPAA and GDPR. Our method helps in detecting whether such a summarization model can inadvertently leak specific details from its training data. **By evaluating these risks beforehand, appropriate privacy defense mechanisms can be implemented prior to model deployment.**
>
> **References**\
> [1] Krishna, Kundan, et al. "Generating SOAP notes from doctor-patient conversations using modular summarization techniques." ACL 2020.\
> [2] Jain, Raghav, et al. "A survey on medical document summarization." arXiv preprint arXiv 2022.

---

### Official Review · Reviewer_Eh5X · 2023-08-06

**Soundness:** 3

**Excitement:**

4: Strong: This paper deepens the understanding of some phenomenon or lowers the barriers to an existing research direction.

**Paper Topic And Main Contributions:**

This paper presents approaches to determine Membership Inference (MI) in the context of text summarization. MI is a form of attack on LLMs to determine the risk of exposing training data. They present studies in two scenarios: (a) when the attacker has access to the reference summary using one or more API calls and (b) when the attacker has access to only the model and not the reference summary. They show that current summarization models are at risk of exposing data membership even in cases where the reference summary is not available.

Strengths:
They perform extensive experiments across three datasets to determine MI attack in summarization models. The results from the paper seem generalizable.
Their ablation studies highlight the impact of overfitting, data size and model architcture.
They also discuss the plausible defense strategies against such attacks.
The paper is written clearly and is easy to comprehend.

Weaknesses:
They employ BART-Base and Flan-T5 models for summarization. Perhaps they could use more powerful models.
(not really a weakness?) They tried three augmentation strategies: Word Synonym, Sentence swapping and Back Translation. However, these strategies seem similar to each other. Perhaps they could have employed slightly different strategies such as attempting to summarize from overlapping segments of the input document, etc.
There are some typos in the paper which can be improved. For example, line 87 contains the word ‘lunch’ where as it should have been ‘launch’

**Reasons To Accept:**

- Novel task formulation

- Extensive experimentation and ablation studies highlighting the impact of various factors

- Variety of datasets employed in the study.

- Propose plausible defense mechanisms.

**Reasons To Reject:**

- Not the most up to date models employed.

**Reproducibility:**

3: Could reproduce the results with some difficulty. The settings of parameters are underspecified or subjectively determined; the training/evaluation data are not widely available.

**Reviewer Confidence:**

3: Pretty sure, but there's a chance I missed something. Although I have a good feel for this area in general, I did not carefully check the paper's details, e.g., the math, experimental design, or novelty.

---

> ### Author Rebuttal · Authors · 2023-08-29
>
> **Q1:** The author(s) employ BART-base and Flan-T5 models for summarization. Perhaps they could use more powerful models.
>
> **R1:** Thank you for your constructive suggestion.
>
> - We selected Flan-T5 and BART-base models as **they are commonly employed as foundational models in summarization research and demonstrate commendable performance.** Thus, we consider these models to be representative when assessing the membership inference attack risk for summarization tasks.
> - According to your suggestion, we tested our method on the recent large language model Vicuna-7B [1], recognized as one of the leading open-source models. We fine-tuned Vicuna on the Samsum and CNNDM datasets and applied the baseline and document augmentation attacks, utilizing MLP as the classifier. For augmentation, we selected the sentence swap method, which proved the most effective for membership inference attacks in our study. **As Table 1 depicts, Vicuna-7B is vulnerable to membership inference attacks,** evidenced by an attack AUC greater than 50%. Additionally, the TPR_0.1\% results indicate that the attack can identify specific training data with high accuracy and confidence. We will add these results in the Appendix.
>
> **Table 1: MI Attack Performance on Vicuna-7B Model**
> |MI Attack $\downarrow$| Dataset $\downarrow$|ACC|AUC|TPR_0.1\%|
> |:---:|:---:|:---:|:---:|:---:|
> |Baseline Attack| SAMsum | 61.87 | 65.43 | 1.23 |
> || CNNDM| 52.15 | 56.86| 0.29|
> |Document Augmentation Attack | SAMsum |62.17 | 65.79 |1.50 |
> ||CNNDM | 53.01 | 57.85 | 0.81 |
>
> **Q2:** They tried three augmentation strategies: Word Synonym, Sentence swapping, and Back Translation. However, these strategies seem similar to each other. Perhaps they could have employed slightly different strategies, such as attempting to summarize from overlapping segments of the input document, etc.
>
> **R2:** Thanks for the insightful comment.
> - We have chosen three distinct augmentation strategies, **each offering a unique perspective on linguistic perturbation: lexical, structural, and contextual.** The Word Synonym technique targets lexical variations, Sentence Swapping addresses the structural sequencing of information, while Back Translation introduces nuanced phrasing differences. Collectively, we believe these methods encapsulate the majority of perturbations relevant to input documents.
> - Although there are other potential augmentation strategies, including summarizing from overlapping document segments, **it may require a different training approach and modification of the inference strategy.** Our current focus is on document perturbations, as these are most practical for countering real-world black-box attacks. However, we appreciate the suggestion and are open to exploring alternative strategies in future research.
>
>
> **Q3:** There are some typos in the paper which can be improved. For example, line 87 contains the word ‘lunch’ where as it should have been ‘launch’.
>
> **R3:** Thank the reviewer for pointing out the typos. We have refined this typo in our updated version and will thoroughly review the paper.
>
> **Q4:** This paper presents an approach to denoise weakly supervised labels. They focus on enhancing the quality of weak labels using k-fold cross-validation.
>
> **R4:** Thank you for taking the time to review our paper. We appreciate your comments, but it seems there may be a misunderstanding. The focus of our work is not on denoising weakly supervised labels using k-fold cross-validation. We wonder if this particular feedback might have been intended for another paper. We would be happy to address any concerns that are directly related to our research.
>
>
> **References**
>
> [1] Chiang, Wei-Lin, et al. "Vicuna: An open-source chatbot impressing gpt-4 with 90%* chatgpt quality." See https://vicuna. lmsys. org (accessed 14 April 2023) (2023).

---

### Meta-Review · Area_Chair_hwrH · 2023-09-17

**Recommendation:** 4

**Metareview:**

The paper presents an interesting, basic and effective method to identify whether models enclose information from training data. The method can work on a blackbox model and provides extensive experiments on multiple dataset.

Apart from minor improvements in the writing, reviewers point out that the method is tested on small models and it could be the case that it works less well for larger ones, so this is a nice first step, and that the choice of summarisation where most other papers investigate machine learning, it is harder to compare to other methods.

Overall, the paper presents a nice first step that can be explored further (does it also work with larger models, how does it compare to other methods when applied to ML) in the future.

---

### Decision · Program_Chairs · 2023-10-07

**Decision:**

Accept-Findings

**Comment:**

The paper presents an interesting, basic and effective method to identify whether models enclose information from training data. The method can work on a blackbox model and provides extensive experiments on multiple dataset.

Apart from minor improvements in the writing, reviewers point out that the method is tested on small models and it could be the case that it works less well for larger ones, so this is a nice first step, and that the choice of summarisation where most other papers investigate machine learning, it is harder to compare to other methods.

Overall, the paper presents a nice first step that can be explored further (does it also work with larger models, how does it compare to other methods when applied to ML) in the future.